# Zinc Administration and Improved Serum Markers of Hepatic Fibrosis in Patients with Autoimmune Hepatitis

**DOI:** 10.3390/jcm10112465

**Published:** 2021-06-02

**Authors:** Kei Moriya, Norihisa Nishimura, Tadashi Namisaki, Hiroaki Takaya, Yasuhiko Sawada, Hideto Kawaratani, Kosuke Kaji, Naotaka Shimozato, Shinya Sato, Masanori Furukawa, Akitoshi Douhara, Takemi Akahane, Akira Mitoro, Junichi Yamao, Hitoshi Yoshiji

**Affiliations:** 1Department of Gastroenterology and Hepatology, Nara Medical University, 840 Shijo-cho, Kashihara, Nara 634-8522, Japan; nishimuran@naramed-u.ac.jp (N.N.); tadashin@naramed-u.ac.jp (T.N.); htky@naramed-u.ac.jp (H.T.); yasuhiko@naramed-u.ac.jp (Y.S.); kawara@naramed-u.ac.jp (H.K.); kajik@naramed-u.ac.jp (K.K.); shimozato@naramed-u.ac.jp (N.S.); shinyasato@naramed-u.ac.jp (S.S.); aki-do@hotmail.co.jp (A.D.); stakemi@naramed-u.ac.jp (T.A.); mitoroak@naramed-u.ac.jp (A.M.); yoshijih@naramed-u.ac.jp (H.Y.); 2Department of Endoscopy, Nara Medical University, 840 Shijo-cho, Kashihara, Nara 634-8522, Japan; furukawa@naramed-u.ac.jp (M.F.); juny3126@naramed-u.ac.jp (J.Y.)

**Keywords:** autoimmune hepatitis, liver fibrosis, matrix metalloproteinase, serum zinc

## Abstract

Aim: The aim of the present study is to investigate the effect of long-term zinc supplementation, which is important for the activation of various enzymes that contribute to antioxidant and antifibrotic activities, on the improvement of serum fibrotic markers in patients with autoimmune hepatitis (AIH). Methods: A total of 38 patients with AIH under regular treatment at our hospital who provided their consent for being treated with polaprezinc (75 mg twice daily) were included and classified into 2 groups: the patients with zinc elevation (*n* = 27) and the patients without zinc elevation (*n* = 11). Serum biomarker of fibrosis, protein expression levels of matrix metalloproteinases (MMPs), and their inhibitors (TIMPs) were evaluated. Results: A significant difference was found between the variability of serum procollagen type Ⅲ and collagen type Ⅳ-7S between the 2 groups before and after zinc administration for more than 24 months (*p* = 0.043 and *p* = 0.049). In the patients with zinc elevation, no significant changes were found in collagenase (MMP-1 and MMP-13) before and after zinc administration, whereas a significant increase in the expression of gelatinase (MMP-2 and MMP-9) was found after administration (*p* = 0.021 and *p* = 0.005). As for the relative ratio of MMPs to TIMPs, only MMP-9 to TIMP-1 showed a significant increase (*p* = 0.004). Conclusions: Long-term treatment with polaprezinc has been demonstrated to safely improve serum fibrosis indices through increases in MMP-2/-9 and MMP-9/TIMP-1 and is expected to be well combined with direct antifibrotic therapies such as molecularly targeted agents.

## 1. Introduction

An increasing trend of prevalence of autoimmune hepatitis (AIH) was found according to the nationwide epidemiologic survey [1]. However, the etiology of AIH still remains largely unknown, and no therapeutic agents that can “cure” this disease have been developed [2]. For a tentative treatment, as indicated in the current clinical practice guidelines of the European Association for the Study of the Liver and the American Association for the Study of Liver Diseases, corticosteroid is definitely the first-line treatment and is recommended for a patient with AIH [3,4]. Although in a multi-center prospective cohort study, Yoshizawa et al. reported that long-term outcomes of patients with AIH were comparable with those of the general population; almost all of the patients were intensively treated with prednisolone [5]. Long-term use of prednisolone has been widely known to be associated with various kinds of adverse effects with a high occurrence ratio including the increased susceptibility to infection such as *Pneumocystis jirovecii* and coronavirus disease 2019 [6,7,8].

In such a situation, clinicians sometimes encounter a case of AIH whose transaminase level remains within the normal limit, but its histological finding reveals mild active hepatitis. Dhaliwal et al. [9] reported that patients with AIH were supposed to have a relatively poor prognosis. Similar to other kinds of chronic liver disease such as nonalcoholic steatohepatitis and viral hepatitis [10,11], the progression of fibrosis is an important factor to predict the long-term prognosis in patients with chronic liver disease, including AIH [12,13,14]. In this point, keeping the pathological activity of AIH as quiet as possible and inhibiting its fibrosis as definitely as possible are the most important actions for patients with AIH to reach good prognosis.

Next to iron, zinc is the second largest mineral element stored in the human body and has a pivotal role as a micronutrient for antioxidant, antiinflammatory, and antiapoptotic effects [15,16]. Zinc also plays an important role as a cofactor of some enzymes involved in collagen synthesis [17,18]. Collagenase is a zinc metalloenzyme and zinc deficiency causes a decline in the activity of collagenase that results in liver fibrosis [19]. Zhou et al. reported that zinc has cytoprotective activities that protect hepatocytes from oxidative stress. In an animal model, zinc deficiency enhanced sensitivity to drug-induced hepatotoxicity and zinc supplementation suppressed the collagen synthesis in hepatic stellate cells (HSCs) [20,21]. Polaprezinc, composed of elemental zinc and L-carnosine, is known to have an effect of tissue repair, active oxygen removal, and anti-inflammatory properties, and has been available in clinical use for peptic ulcer in Japan. This prospective clinical study aimed to elucidate the inhibitory effect of polaprezinc on the progression of fibrosis in patients with AIH non-invasively through molecular mechanisms. To the end, we evaluated the changes of serum fibrosis indices before and after the zinc administration.

## 2. Methods

### 2.1. Patients

Of the 79 patients with histologically diagnosed AIH based on the revised diagnostic scoring system of the International Autoimmune Hepatitis Group (IAIHG) in 1999 and continuous treatment at the Nara Medical University Hospital, 49 patients were enrolled in this prospective study between September 2015 and March 2017 with written informed consent from each individual. The inclusion criteria were as follows: (i) patients who have successfully achieved clinical remission of AIH (in which serum levels of transaminase continuously stayed within normal limits for at least six months), and (ii) with continuous administration of polaprezinc for over two years. The exclusion criteria were as follows: (i) positive for hepatitis C virus (HCV), hepatitis B virus (HBV), and/or human immunodeficiency virus (HIV); (ii) with some liver disease; (iii) under the administration of hepatotoxic drugs; and (iv) with decompensated cirrhosis, severe cytopenia, renal failure, heart failure, and pregnant or lactating women. For zinc administration, polaprezinc (75 mg twice daily) was prescribed to study participants. In addition, we excluded 11 patients who took polaprezinc for <24 months after the study enrollment. All of them discontinued polaprezinc within a year (mean ± standard deviation (SD), 4.9 ± 3.6 months) because of their intentions without any serious reason or discontinuation of medical maintenance owing to reasons such as moving (Appendix A). A patient quit polaprezinc because of his mild diarrhea, but he quickly recovered after that. Finally, the remaining 38 patients who successfully took polaprezinc for >24 months were divided into two groups: the patients with zinc elevation and the patients without zinc elevation (Figure 1).

The patients with zinc elevation (*n* = 27) were those whose serum zinc level at 12 months after the beginning of administration was >80 μg/dL (the normal lower limit in Japan) and was also increased to >5% during this period. The patients without zinc elevation (*n* = 11) were those whose serum zinc level after the supplementation was either ≤80 μg/dL or increased to ≤5% during this period.

### 2.2. Laboratory Assessments

All patients underwent a routine laboratory assessment in the hospital, which included complete blood count, a general biochemistry test, and coagulation test. Serum levels of procollagen type Ⅲ, type Ⅳ collagen 7S, and hyaluronic acid were measured by external clinical examination facilities with chemiluminescent immunoassay, chemiluminescent enzyme immunoassay, and latex immunoagglutination assay, respectively. Serum levels of human matrix metalloproteinase-1 (MMP-1), metalloproteinase-2 (MMP-2), metalloproteinase-9 (MMP-9), and metalloproteinase-13 (MMP-13) were measured using the following enzyme-linked immunosorbent assay (ELISA) kits: human MMP-1 ELISA kit #ELH-MMP1 (RayBio^®^, Norcross, GA, USA), human MMP-2 ELISA kit #KE00077, human MMP-9 ELISA kit #KE00164, and human MMP-13 ELISA kit #KE00078 (Proteintech^®^, Rosemont, IL, USA). Serum levels of the inhibitor of MMPs, TIMP-1, and TIMP-2 were measured using the following ELISA kits according to the manufacturer’s instructions: human TIMP-1 ELISA kit #DTM100 and human TIMP-2 ELISA kit #DTM200 (R&D Systems^®^, Minneapolis, MN, USA).

### 2.3. Histological Assessments

The biopsy samples were stained using hematoxylin/eosin and Azan methods. Liver fibrosis was staged based on the METAVIR score (F0, no fibrosis; F1, portal fibrosis without septa; F2, portal fibrosis with few septa; F3, numerous septa without cirrhosis; F4, cirrhosis). F0 to F2 was considered as non-fibrosis to mild fibrosis, whereas F3 to F4 was considered as advanced fibrosis.

### 2.4. Statistical Analyses and Ethical Issues

The numerical variables were expressed as mean ± SD. The chi-square test, Mann-Whitney *U* test, and Wilcoxon signed-rank test were used to compare patient characteristics between the groups. Correlation was assessed by using Spearman’s rank correlation coefficients. *p* < 0.05 was considered statistically significant. JMP version 14.3 software (SAS Institute Inc., Cary, NC, USA) was used for statistical analyses.

This study was approved by the Ethics Committee of the Nara Medical University Hospital (approval #15-003) and was conducted according to the ethical principles in the Japanese ethical guidelines for epidemiologic research (https://www.mhlw.go.jp/stf/seisakunitsuite/bunya/hokabunya/kenkyujigyou/i-kenkyu/index.html (accessed on 1 December 2020)). This study was conducted according to the Declaration of Helsinki. This study protocol was registered as a clinical trial (UMIN000022959, https://upload.umin.ac.jp/ (accessed on 13 June 2016)), and a written informed consent was obtained from all patients.

## 3. Results

The clinical profiles of patients in each group before starting zinc supplementation are presented in Table 1. No significant differences were found in age, sex, body weights, and the degree of hepatic fibrosis. There were no significant differences in the basal serum levels of zinc, procollagen III and collagen IV-7S. All study participants were well treated with ursodeoxycholic acids and/or prednisolone, and their transaminase levels were within the normal range. Although the hepatitis activities of patients in both groups were modest, their serum zinc levels seemed to be insufficient.

Before zinc supplementation, serum albumin levels, which generally predict the overall survival of patients with liver diseases [22], correlated with serum zinc levels of patients with AIH in this study (*p* < 0.05) (Figure 2A). Serum zinc levels also inversely correlated with some liver fibrotic markers, such as serum procollagen type Ⅲ, collagen type Ⅳ-7S, and hyaluronic acid (*p* < 0.05) (Figure 2B–D, respectively).

Serum zinc concentration was increased to >50% at 12 or 24 months after continuous zinc supplementation in the patients with zinc elevation (before, 70.8 ± 10.9 μg/dL; 12 months, 109.9 ± 35.5 μg/dL (*p* < 0.001); 24 months, 105.1 ± 38.0 μg/dL (*p* < 0.001)) (Table 2). The serum ferritin levels (before, 84.4 ± 95.6 ng/mL; 12 months, 56.8 ± 50.8 ng/mL (*p* = 0.19); 24 months, 54.3 ± 42.0 ng/mL (*p* = 0.20)) in the patients with zinc elevation and their serum copper levels (before, 121.8 ± 27.7 μg/dL; 12 months, 107.8 ± 17.7 μg/dL (*p* = 0.34) 24 months, 113.8 ± 20.6 μg/dL (*p* = 0.25)) were not affected and no clinical adverse event such as anemia and neuropathy was observed in this study (Table 2). Although there was no change in the transaminase levels during the entire observation period, the serum levels of procollagen type Ⅲ and collagen type Ⅳ-7S indicated a tendency to decrease at 12 months after zinc supplementation (*p* = 0.06 and *p* = 0.07, respectively). In contrast, in the patients without zinc elevation, serum zinc concentration continuously remained at the baseline level through the course, although their transaminase and the other biliary enzyme levels never changed (Appendix A). To our surprise, the levels of serum procollagen type Ⅲ and collagen type Ⅳ-7S indicated a gradient increase, but there was no significance because of the small number of cases.

Based on these findings, we compared the changes of procollagen type Ⅲ level in the patients with zinc elevation and the patients without zinc elevation at 12 months and 24 months after zinc supplementation. There were significant differences between the patients with zinc elevation and the patients without zinc elevation in terms of the difference in serum procollagen III level at 24 months after zinc supplementation (*p* = 0.043) (Figure 3A). Moreover, in terms of serum collagen type Ⅳ-7S level, significant differences between the patients with zinc elevation and the patients without zinc elevation were also found at 12 months and 24 months after zinc supplementation (*p* = 0.050 and *p* = 0.049, respectively) (Figure 3B). 

Finally, we also investigated the protein levels of serum collagenase (MMP-1 and MMP-13), gelatinase A (MMP-2), and gelatinase B (MMP-9). In addition, the serum levels of the inhibitor of MMPs, such as TIMP-1 and TIMP-2, were also examined. In the patients with zinc elevation, the levels of these gelatinases were significantly increased after zinc supplementation for >24 months, whereas those of collagenase were not (Figure 4A). The serum level of TIMP-2 was significantly augmented after zinc supplementation, but no significant change was found in TIMP-1 level (Figure 4B). In terms of the ratio of each MMP and its related inhibitor, MMP-9/TIMP-1 was significantly up-regulated. However, there were no significant changes in other ratios, such as MMP-1/TIMP-1, MMP-13/TIMP-1, and MMP-2/TIMP-2 (Figure 4C).

## 4. Discussion

Therapeutic strategies for liver fibrosis include indirect approaches such as the use of nucleic acid analogs and direct-acting antiviral agents, which are first and foremost aimed at eliminating disease factors. In contrast, recent experiments using cell lineage tracing technology and reporter mice have shown that HSCs play a much larger role in collagen production than myeloid cells and myofibroblasts, regardless of the type of liver fibrosis [23,24,25]. Therefore, in recent years, direct antifibrotic drugs, such as molecularly targeted drugs, small-molecule compounds, antibody drugs, and nucleic acid drugs, have been developed, many of which exert their antifibrotic effects by acting on molecular mechanisms closely related to liver fibrosis, such as inhibition of HSC activation [26,27,28], inhibition of the response cascade after HSC activation [29,30], or induction of apoptosis of activated HSC [31,32,33,34,35,36,37]. These include research and development targeting intrinsic interferons [34,35], but zinc was recently reported to specifically inhibit proinflammatory cytokines and IFNλ3 signaling to improve fibrosis [38]. In their study, Read et al. found that elevated zinc levels in liver tissue induced metallothionein expression and suppressed IFNλ3 expression, resulting in the attenuation of gene expression of interferon-stimulated genes and inflammatory cytokines, thereby suppressing antiviral activity and its immune response and ultimately negatively regulating liver fibrosis.

Zinc is essential for the maintenance of life activities and is an essential trace element, although its content in the body is only 2–3 g [39]. The fact that 10% of proteins encoded in the human genome contain a zinc-binding motif means that a very large number of proteins bind to zinc [40], which suggests the importance of zinc’s signaling function inside and outside the cell [41]. In fact, there have been several reports of improvement in liver function with the administration of zinc products, mainly in patients with chronic hepatitis C and cirrhosis [42,43,44,45,46]. Regarding the association between serum zinc concentrations and long-term prognosis in patients with chronic liver disease, in a multicenter, long-term observational study of cirrhotic patients (mean follow-up period, 3 years), Shigefuku et al. found that low serum zinc concentrations (<55 μg/dL) at enrollment were an independent risk factor for liver carcinogenesis [47].

Liver fibrosis is a common pathological process characterized by an accumulation of the extracellular matrix (ECM), which is a kind of tissue remodeling triggered by a consequence of an imbalance between the enhanced ECM synthesis and reduced degradation of connective tissue proteins reflecting the dysregulation of several pathways, including MMPs and tissue inhibitors of MMPs (TIMPs) [48]. MMPs are a large family of zinc-dependent enzymes that degrade the components of ECM [49], whereas TIMPs are specific endogenous inhibitors that bind to MMPs and block them from ECM components. The most potent MMPs are collagenases such as MMP-1, MMP-8, and MMP-13 [50], and the activity of MMPs is highly regulated by the level of gene expression, activation of latent pro-MMPs to active enzymes [51], and TIMPs that form stable, noncovalent complexes with active MMPs [52].

Furthermore, MMPs are known as representative zinc-containing enzymes, and there are strong expectations regarding the effect of zinc preparations on fibrinolysis. The inhibition of hepatic fibrosis by zinc administration would be based on its effects of MMPs involved in collagen synthesis and degradation and its ability to control the function of procollagen and collagen-producing hepatic stellate cells by inhibiting oxidative stress, inflammation, and apoptosis in the liver. Apart from research reagents, the available zinc-containing preparations are currently limited to zinc acetate and polaprezinc. Although there have been some reports on the sedative or prognostic effects of continuous administration of these products [42,43,44,45,46], studies on their effects on the inhibition of hepatic fibrosis have been limited to basic studies, mainly in rodents [53,54,55], and there have been few reports on the changes in the expression of MMPs or TIMPs in particular. In this regard, in a small but prospective pilot study in a small number of patients with highly advanced chronic liver disease, Takahashi et al. [56] found no significant change in MMPs but a significant decrease in TIMP-1 after polaprezinc administration, suggesting that the decrease in TIMP-1 might be important in the improvement of fibrosis. In contrast, our results indicated a significant increase in MMP-2 and MMP-9 after polaprezinc treatment. In addition, TIMP-2 showed a significant increase after zinc administration, whereas TIMP-1 did not change with significance. Subsequently, MMP-9/TIMP-1 was also significantly increased, whereas MMP-2/TIMP-2 was not affected. Based on these results, not only the increase in MMPs but also the increase in MMP-9/TIMP-1 can be responsible for the improvement of fibrosis. The reason for the significant increase in MMP-2 and MMP-9 after zinc administration in our study is that the disease activity of AIH has already been in clinical remission by corticosteroids and or ursodeoxycholic acid, and the transaminases of the subject patients remained within the normal range during the entire observation period. In this aspect, during the maintenance of remission, MMPs were relatively lower (closer to those of healthy individuals) than during the active phase of the disease, which may facilitate confirmation of the increase in MMPs after zinc administration, which may be a major difference between our study and the other previous studies, which were conducted in patients with viral active hepatitis or liver cirrhosis and in animal models with drug-induced liver injury.

In a family of MMPs, Latronico et al. [57] reported that serum production levels of MMP-2 and MMP-9 were definitely higher than the other MMPs such as MMP-1, MMP-3, MMP-8, and MMP-10, and their levels significantly increased in patients with hepatitis compared with healthy subjects. In that study, serum TIMP-1 levels were correlated with liver stiffness, which reflected the degree of fibrosis, and the result was compatible with the other previous report [58]. Giannelli G et al. [59] reported that the ratio of serum MMP-2/TIMP-2 was definitely lower in cirrhotic patients than healthy subjects, whereas serum MMP-2 levels in cirrhotic patients were similar to those of healthy subjects. In contrast, Watanabe et al. [60] reported that serum MMP-2 levels increased in parallel with the progression of chronic liver disease, and there was a positive correlation between the collagen type Ⅳ and MMP-2 levels. Subsequently, an imbalance between MMP-2 and the TIMP-2 might be also responsible for the degradation of ECM components (Figure 5A). In terms of the increase of TIMP-2 after zinc administration in our study, we believe that this is probably an equilibrium response (positive feedback) associated with an increased level of MMP-2, which has a binding affinity with TIMP-2. That is, even though both MMP-2 and TIMP-2 were increased after zinc administration, if MMP-2/TIMP-2 was not decreased, it would be acceptable that the vector in the direction of fibrosis was pointing to amelioration (Figure 5B).

Regarding the safety of zinc preparations, the permissible maximum dose of oral zinc intake for a short period was reported to be approximately 170 mg per day in healthy adult subjects, and this was almost equivalent to 10 times volume of the average intake. In this study, patients took 150 mg of polaprezinc (containing 34 mg of elemental zinc) every day. In contrast, there have been reports of copper deficiency with the long-term administration of polaprezinc and zinc acetate [61,62,63], as well as reports that caution should be exercised when using zinc acetate rather than polaprezinc [64]. In this study involving polaprezinc, there were no cases of serum copper below baseline levels and no adverse physical findings, such as anemia or neurological symptoms, were observed, including in preexcluded subjects because of their insufficiency of taking polaprezinc. According to the medical guidelines on zinc deficiency [65], many cases of serum copper levels of <10 μg/dL and serum zinc levels of 190–250 μg/dL at the onset of copper deficiency were common; hence, copper deficiency should be noted when serum copper levels are 20–30 μg/dL and serum zinc levels are >200 μg/dL. In addition, although rare, zinc administration can cause inhibition of iron absorption in the intestinal tract, leading to iron deficiency, and therefore, iron deficiency should be noted in the same way as copper [66].

This study has some limitations: (1) a relatively small number of patients were enrolled in this single-center study, (2) tissue sampling by liver biopsy after zinc administration is not sufficiently available and is poorly supported by histopathology, (3) there are relatively few cases of advanced fibrosis, and (4) the study was limited to Japanese patients with AIH, who are generally considered to have a good prednisolone response. Certainly, notwithstanding these limitations, it is significant that our results demonstrate that long-term continuous treatment with zinc preparations for >2 years improves fibrotic markers through increases in MMPs and MMP/TIMPs and that it is safe. Azathioprine, which, along with prednisolone, is the first-line drug for AIH, has a chelating effect that binds metals, and is excreted in the urine after binding to zinc in the body. Considering the metabolism of azathioprine, the possibility of zinc deficiency owing to increased urinary zinc excretion in long-term treatment cannot be ruled out. In this context, it may be worthwhile to use zinc in combination with AIH treatment.

In conclusion, long-term zinc administration for AIH could be a safe and effective antifibrotic treatment. In addition, a zinc acetate preparation with a higher zinc content than polaprezinc has recently been approved for insurance use in Japan, and it is expected to be combined with direct antifibrotic drugs such as molecularly targeted drugs, small-molecule compounds, antibody drugs, and nucleic acid drugs that are currently being developed.

## Figures and Tables

**Figure 1 jcm-10-02465-f001:**
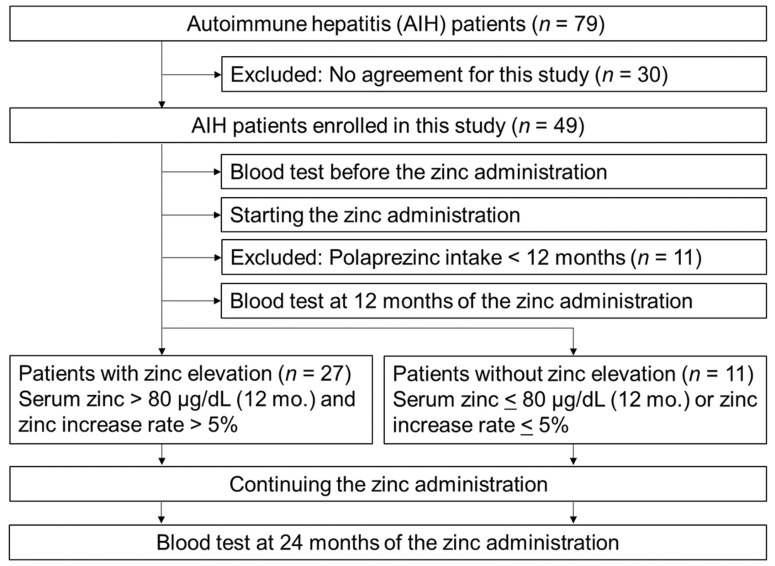
Patient flowchart. A total of 49 patients with AIH were initially included in analysis, 11 of whom were excluded because they withdrew polaprezinc intake within 12 months; 38 patients treated with polaprezinc for >24 months were enrolled and were divided into two groups based on serum zinc level and its increase rate.

**Figure 2 jcm-10-02465-f002:**
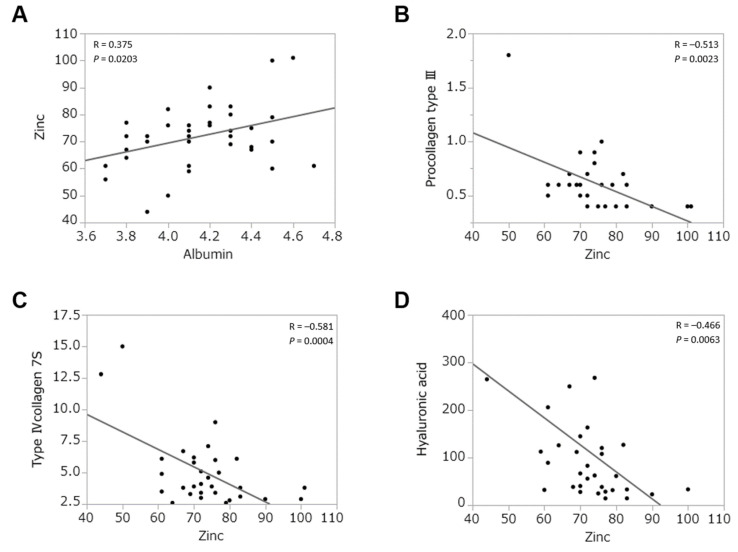
Relations between basal serum zinc level and biochemical markers. (**A**) Relation between zinc and albumin (*p* = 0.020). (**B**) Relation between zinc and procollagen type Ⅲ (*p* = 0.002). (**C**) Relation between zinc and collagen type Ⅳ-7S (*p* < 0.001). (**D**) Relation between zinc and hyaluronic acid (*p* = 0.006).

**Figure 3 jcm-10-02465-f003:**
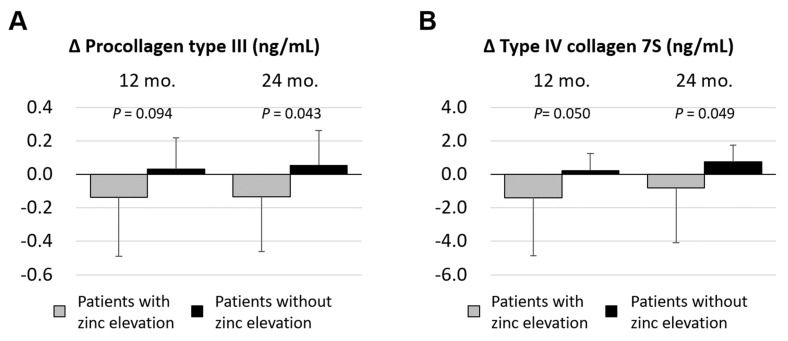
Longitudinal changes of serum fibrotic markers beyond zinc supplementation. (**A**) Changes of serum procollagen type III level after zinc administration of 12 and 24 months (*p* = 0.094 and *p* = 0.043). (**B**) Changes of serum collagen type IV-7S after zinc administration of 12 and 24 months (*p* = 0.050 and *p* = 0.049).

**Figure 4 jcm-10-02465-f004:**
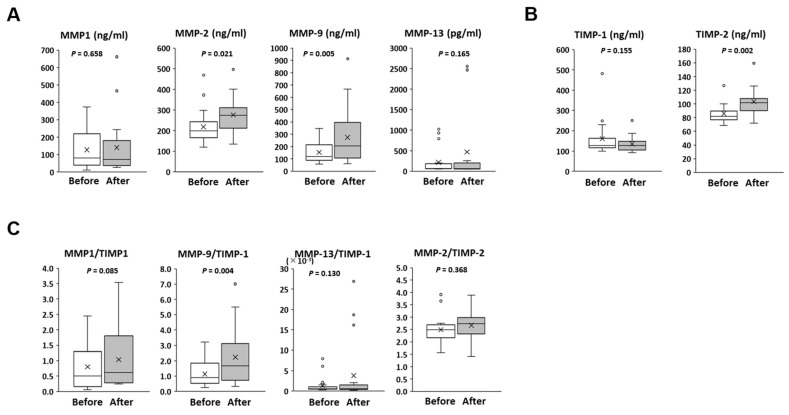
Longitudinal changes of serum matrix metalloproteinase (MMP) and its inhibitor (TIMP) in the patients with zinc elevation. (**A**) Longitudinal changes of serum collagenases level (MMP-1 and MMP-13) and gelatinase level (MMP-2 and MMP-9). (**B**) Longitudinal changes of serum inhibitor of MMP (TIMP-1 and TIMP-2). (**C**) Relative ratios of each MMP and its inhibitor.

**Figure 5 jcm-10-02465-f005:**
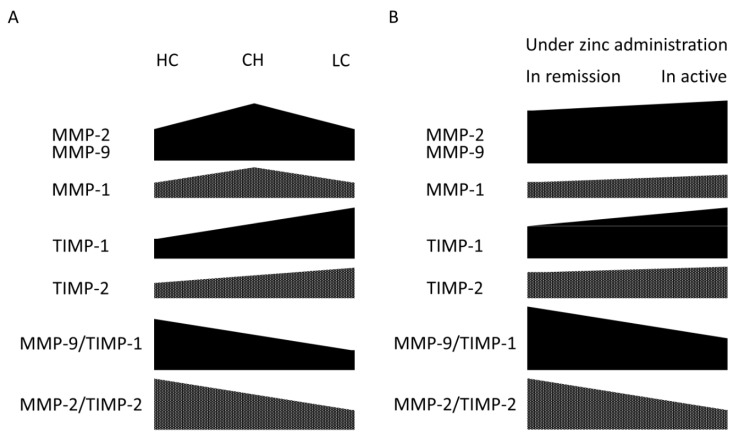
Scheme of the serum levels of matrix metalloproteinase (MMP) and its inhibitor (TIMP) in the course of liver disease progression and mechanistical working-hypothesis of the changes in MMP and TIMP levels under zinc administration. (**A**) Serum levels of MMPs and TIMPs in healthy carriers (HC), patients with chronic hepatitis (CH), and patients with liver cirrhosis (LC). (**B**) Serum levels of MMPs and TIMPs in patients with chronic liver disease in remission and in active phase.

**Table 1 jcm-10-02465-t001:** Clinical profiles of patients with autoimmune hepatitis before zinc supplementation.

	Patients with Zinc Elevation			Patients without Zinc Elevation			*p*
	(n = 27)			(n = 11)			
Age (years)	65.0	±	11.0	69.2	±	13.9	0.38
Sex							1.00
Male	7.4	%	(2)	9.1	%	(1)	
Female	92.6	%	(25)	90.9	%	(10)	
Body Weights (kg)	54.0	±	9.8	53.0	±	10.1	0.79
Body Mass Index (kg/m^2^)	23.0	±	3.8	23.5	±	4.2	0.74
Fibrosis							0.81
F0	3.7	%	(1)	0.0	%	(0)	
F1	44.4	%	(12)	54.5	%	(6)	
F2	29.6	%	(8)	27.3	%	(3)	
F3	14.8	%	(4)	9.1	%	(1)	
F4 (Liver Cirrhosis)	7.4	%	(2)	9.1	%	(1)	
AIH type 1	100	%	(27)	100.0	%	(11)	1.00
AIH type 2	0	%	(0)	0.0	%	(0)	1.00
Morbidity periods (years)	4	±	3.4	5.3	±	3.1	0.30
Blood test							
Zinc (µg/dL)	70.8	±	10.9	74.4	±	13.0	0.44
AST (U/L)	24.6	±	9.2	29.4	±	13.6	0.31
ALT (U/L)	18.1	±	10.1	20.3	±	9.5	0.54
ALP (U/L)	233	±	100	308	±	152	0.15
gamma GT (U/L)	38.9	±	61.2	69.5	±	109	0.40
Total bilirubin (mg/dL)	0.79	±	0.45	0.85	±	0.29	0.65
Albumin (g/dL)	4.16	±	0.26	4.11	±	0.27	0.59
HDL-cholesterol (mg/dL)	67.6	±	22.1	71.3	±	19.5	0.61
Ferritin (ng/mL)	84.4	±	95.6	97.7	±	77.1	0.66
Copper (µg/dL)	121.8	±	27.7	130.3	±	23.1	0.38
IgG (mg/dL)	1389	±	327	1495	±	488	0.52
Hyaluronic acid (ng/mL)	92.2	±	78.2	168	±	187	0.25
Pro-collagen type III (ng/mL)	0.65	±	0.30	0.55	±	0.17	0.25
Type IV collagen 7S (ng/mL)	5.45	±	3.13	4.09	±	1.24	0.08
Platelets (/µL)	19.6	±	5.7	18.9	±	5.4	0.72
Prothrombin time (%)	95.4	±	19.7	96.6	±	16.6	0.84
Treatments							
PSL populations (%)	51.9	%	(14)	36.4	%	(4)	0.48
PSL dose (mg)	5.8	±	1.8	6.3	±	1.9	0.48
UDCA populations (%)	88.9	%	(24)	90.9	%	(10)	1.00
UDCA dose (mg)	600	±	177	630	±	95	0.53
Azathioprine populations (%)	11.0	%	(3)	9.1	%	(1)	0.85
Azathioprine dose (mg)	100	±	0	100	±	0	1.00

**Table 2 jcm-10-02465-t002:** Clinical parameters of the patients with zinc elevation before and after zinc supplementation.

Patients with Zinc Elevation (n = 27)	Before			After 12 Months				After 24 Months			
	Mean		SD	Mean		SD	*p*	Mean		SD	*p*
Zinc (µg/dL)	70.8	±	10.9	109.9	±	35.5	<0.001	105.1	±	38.0	<0.001
AST (U/L)	24.6	±	9.2	24.4	±	11.1	0.93	27.2	±	17.8	0.51
ALT (U/L)	18.1	±	10.1	17.0	±	9.3	0.67	19.0	±	13.4	0.79
ALP (U/L)	232.8	±	100.0	275.2	±	279.8	0.46	242.8	±	86.4	0.70
Gamma GT (U/L)	38.9	±	61.2	52.4	±	140.5	0.65	32.3	±	42.0	0.65
Total bilirubin (mg/dL)	0.79	±	0.45	0.78	±	0.42	0.93	0.81	±	0.47	0.88
Albumin (g/dL)	4.16	±	0.26	4.22	±	0.23	0.41	4.21	±	0.18	0.48
HDL-cholesterol (mg/dL)	67.6	±	22.1	64.8	±	19.0	0.63	67.9	±	19.9	0.95
Ferritin (ng/mL)	84.4	±	95.6	56.8	±	50.8	0.19	54.3	±	42.0	0.20
Copper (µg/dL)	121.8	±	27.7	107.8	±	17.7	0.34	113.8	±	20.6	0.25
IgG (mg/dL)	1388.7	±	327.3	1359.3	±	279.5	0.72	1351.8	±	351.7	0.71
Hyaluronic acid (ng/mL)	92.2	±	78.2	65.3	±	53.8	0.16	91.1	±	73.3	0.97
Pro-collagen type III (ng/mL)	0.65	±	0.30	0.52	±	0.16	0.06	0.49	±	0.15	0.06
Type IV collagen 7S (ng/mL)	5.45	±	3.13	4.12	±	1.68	0.07	4.30	±	1.23	0.15
Platelets (/µL)	19.6	±	5.7	20.4	±	5.7	0.61	20.8	±	5.8	0.45
Prothrombin time (%)	95.4	±	19.7	98.7	±	13.4	0.47	95.1	±	13.9	0.96

## Data Availability

The datasets used and/or analyzed during the current study are available from the corresponding author on reasonable request.

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
