# Peer review of "Zinc Administration and Improved Serum Markers of Hepatic Fibrosis in Patients with Autoimmune Hepatitis"

_jcm, 2021, doi:10.3390/jcm10112465_

Round 1

Reviewer 1 Report

Recommendations for authors -I recommend writing the methods section of the Abstract in a more comprehensible and reduced way. I suggest, for example, not to add the exclusion criteria to the Abstract. -I suggest modifying figure 1. I recommend indicating on the graph, using labeled arrows, the time when the first blood test is performed, the zinc supplementation and the time of the next blood tests (12 and 24 months) -In the Results section, I recommend including that "there is no difference between the basal levels of zinc, procollagen III or collagen IV-7S" -In the title of figure 2, I recommend adding the adjective "basal"("relations between basal serum zinc level...")

Author Response

Reviewer 1

I recommend writing the methods section of the Abstract in a more comprehensible and reduced way. I suggest, for example, not to add the exclusion criteria to the Abstract.

First of all, I really appreciate your having reviewed my article and thank you for your valuable comments. According to your suggestion, we omitted the indicated part and rewrote it as “A total of 38 patients with AIH under regular treatment at our hospital who provided their consent for being treated with polaprezinc (75 mg twice daily) were included and classified into 2 groups: the patients with zinc elevation (n=27) and the patients without zinc elevation (n=11).”in the methods section of the abstract.

I suggest modifying figure 1. I recommend indicating on the graph, using labeled arrows, the time when the first blood test is performed, the zinc supplementation and the time of the next blood tests (12 and 24 months).

I really appreciate your suggestion. Please see and check the modified “Figure 1”.

In the Results section, I recommend including that "there is no difference between the basal levels of zinc, procollagen III or collagen IV-7S".

Thank you for your recommendation. I added the sentence “There were no significant differences in the basal serum levels of zinc, procollagen III and collagen IV-7S.” in the result section.

In the title of figure 2, I recommend adding the adjective "basal"("relations between basal serum zinc level...").

Thank you for your suggestion. I added the adjective word “basal” in the title of the Figure 2.

Reviewer 2 Report

In this study the authors showed that zinc administration in AIH patients improves serum fibrosis indices through increases in MMP-2 / -9 and MMP-9 / TIMP-1.
The study has problems to solve:
1. the population studied is very small for adult patients
2. What type of AIH do they suffer from? all adult / elderly patients how many years on average had they been diagnosed?
3. therapies: it is stated that all participants have been taking UDCA and / or steroid, for how long? the steroid at what dosage? anyone with azathioprine?
4. The test used for correlations is missing from the statistics
5. In figure 2 B, C and D the correlations are inverse, why is R expressed in positive? which test was used?
6. Procollagen increases in patients without elevated zinc, as a pro-fibrotic factor, should be explained in the discussion as it is an established marker of fibrosis and why MMPs degrade collagen, then the mechanism linking zinc-MMP and procollagen should be explained.

Author Response

Reviewer 2

In this study the authors showed that zinc administration in AIH patients improves serum fibrosis indices through increases in MMP-2 / -9 and MMP-9 / TIMP-1.

The study has problems to solve:

  1. the population studied is very small for adult patients

First of all, I really appreciate you for having reviewed my current article. As you mentioned, the sample size of this study seemed to be relatively small when compared to another clinical study. However, the targeted disease in this study was autoimmune hepatitis which was one of the uncommon liver diseases worldwide. I would like to set a similar clinical trial as a multicenter study with much more sample size in the future.

  1. What type of AIH do they suffer from? all adult / elderly patients how many years on average had they been diagnosed?

In Japan, AIH classified into the type 2 was very rare. All the AIH patients in this study were classified into the type 1. The average of morbidity periods with AIH in this study was 4.4 years. I added the related information into the Table1. Please see the part of AIH type and morbidity periods.

  1. therapies: it is stated that all participants have been taking UDCA and / or steroid, for how long? the steroid at what dosage? anyone with azathioprine?

Thank you for your comments. The medicated period with UDCA was almost similar to the morbidity period of AIH. Those with steroid or thiopurine were generally shorter. The number of patients treated with azathioprine in each group was very few and the majority of patients in this study was treated with UDCA and / or steroid. I added the clinical data mentioned above in the Table1. Please see the populations and dose of PSL, UDCA, and azathioprine.

  1. The test used for correlations is missing from the statistics.

Thank you for the comment. I added the sentence “Correlation was assessed by using Spearman’s rank correlation coefficients.” in the methods section.

  1. In figure 2 B, C and D the correlations are inverse, why is R expressed in positive? which test was used?

Sorry for my simple mistake. All of the R scores in the Figure 2B, 2C, and 2D should be negative as you kindly taught me, though these absolute scores were the same. Correlation was assessed by using Spearman’s rank correlation coefficients. Please see the corrected Figure 2. I added the sentence “Correlation was assessed by using Spearman’s rank correlation coefficients.” in the methods section.

  1. Procollagen increases in patients without elevated zinc, as a pro-fibrotic factor, should be explained in the discussion as it is an established marker of fibrosis and why MMPs degrade collagen, then the mechanism linking zinc-MMP and procollagen should be explained.

I really appreciate your suggestions. I regret to say, but the serum levels of procollagen type â…¢ in the patient without zinc elevation were not significantly augmented as shown in the Figure 3 and the supplementary Table S2. On the other hand, based on your thoughtful advices, I added the following sentence “The inhibition of hepatic fibrosis by zinc administration would be based on its effects of MMPs involved in collagen synthesis and degradation and its ability to control the function of procollagen and collagen-producing hepatic stellate cells by inhibiting oxidative stress, inflammation, and apoptosis in the liver.” in the discussion section. I thought this sentence actually made it more comprehensible and persuasive. Thanks, again.